# Dual Control Variate for Faster Black-box Variational Inference

## Abstract

Black-box variational inference is a widely-used framework for Bayesian posterior inference, but in some cases suffers from high variance in gradient estimates, harming accuracy and efficiency. This variance comes from two sources of randomness: Data subsampling and Monte Carlo sampling. Whereas existing control variates only address Monte Carlo noise and incremental gradient methods typically only address data subsampling, we propose a new "dual" control variate capable of *jointly* reducing variance from both sources of noise. We confirm that this leads to reduced variance and improved optimization in several real-world applications.

## 1  Introduction

Black-box variational inference (BBVI) [12, 22, 16, 2] has become a popular alternative to Markov Chain Monte Carlo (MCMC) methods. The idea is to posit a variational family and optimize it to be close to the posterior, using only "black-box" access to the target model (evaluations of the density or gradient). This is done by estimating a stochastic gradient of the KL-divergence and deploying it in stochastic optimization. A key advantage of this procedure is that it allows the use of data subsampling in each iteration, which can greatly speed-up optimization with large datasets.

The optimization of BBVI is often described as a doubly-stochastic optimization problem [30, 27] in that BBVI's gradient estimation involves two sources of randomness: Monte Carlo sampling from the variational posterior and data subsampling from the full dataset. Because of the doubly-stochastic nature, one common challenge for BBVI is the variance of the gradient estimates: If this is very high, it forces very small stepsizes, leading to slow optimization convergence [20, 3].

Numerous works have been devoted to reducing the "Monte Carlo" noise that results from drawing samples from the current variational distribution [19, 25, 8, 9, 4]. These methods can typically be seen as creating an approximation of the objective for which the Monte Carlo noise can be integrated exactly, and using this to define a zero-mean random variable, i.e. a control variate, that is negatively correlated with the original gradient estimator. These methods can be used with subsampling by creating approximations for each datum. However, they are only able to reduce Monte Carlo noise for each datum—they do not reduce subsampling noise. This is critical, as subsampling noise is often the dominant source of gradient variance (Sec. 3).

At the same time, for (non-BBVI) optimization problems with *only* subsampling noise, the optimization community has developed incremental gradient methods that "recycle" previous gradient evaluations [26, 28, 13, 6, 7], leading to faster convergence. These methods do not address Monte Carlo noise. In fact, due to the way these methods rely on efficiently maintaining running averages, they cannot typically be applied to doubly-stochastic problems at all.

In this paper, we present a method that *jointly* controls Monte Carlo and subsampling noise in BBVI. The idea is to create approximations of the target for each datum where the Monte Carlo noise

Submitted to 37th Conference on Neural Information Processing Systems (NeurIPS 2023). Do not distribute.

can be integrated exactly. Then, we maintain running averages of the *approximate* gradients, with noise integrated, overcoming the issue of applying incremental gradient ideas to doubly-stochastic problems. The resulting method not only addresses both forms of noise but *interactions* between them as well. We demonstrate through a series of experiments with diagonal Gaussian variational inference on a range of probabilistic models that the method leads to lower variance and significantly faster convergence than existing methods.

## 2 Background: Black-box variational inference

Given a probabilistic model $p(x, z) = \prod_{n=1}^{N} p(x_n \mid z) p(z)$ and observed data $\{x_1, \ldots, x_N\}$, variational inference's goal is to find a tractable distribution $q_w(z)$ with parameters $w$ to approximate the (often intractable) posterior $p(z \mid x)$ over the latent variable $z \in \mathbb{R}^d$. BBVI achieves this by finding the set of parameters $w$ that minimize the KL-divergence from $q_w(z)$ to $p(z \mid x)$, which is equivalent to minimizing the negative Evidence Lower Bound (ELBO)

$$f(w) = -\mathbb{E}_{\mathsf{n}} \mathbb{E}_{q_w(\mathsf{z})} \left[ N \log p(x_{\mathsf{n}} \mid \mathsf{z}) + \log p(\mathsf{z}) \right] - \mathbb{H}(w), \tag{1}$$

where $\mathbb{H}(w)$ denotes the entropy of $q_w$.

Since the inner expectation with respect to $\mathsf{z}$ is typically intractable, BBVI methods rely on stochastic optimization with unbiased gradient estimates. These gradient estimates are typically obtained using the score function method [33] or the reparameterization trick [15, 23, 30]. The latter is often the method of choice, as it usually seems to yield estimators with lower variance. The idea is to define a fixed base distribution $s(\epsilon)$ and a deterministic transformation $\mathcal{T}_w(\epsilon)$ such that for $\epsilon \sim s$, $\mathcal{T}_w(\epsilon)$ is equal in distribution to $q_w$. Then, the objective from Equation (1) can be re-written as

$$f(w) = \mathbb{E}_{\mathsf{n}} \mathbb{E}_{\epsilon} f(w; \mathsf{n}, \epsilon), \quad \text{where} \quad f(w; \mathsf{n}, \epsilon) = -N \log p(x_{\mathsf{n}} \mid \mathcal{T}_w(\epsilon)) - \log p(\mathcal{T}_w(\epsilon)) - \mathbb{H}(w),$$
$$\tag{2}$$

and its gradient can be estimated "naively" by drawing a random $n$ and $\epsilon$, and evaluating

$$g_{\text{naive}}(w; n, \epsilon) = \nabla f(w; n, \epsilon). \tag{3}$$

BBVI has two advantages. First, since it only evaluates $\log p$ (and its gradient) at various points, it can be applied to a diverse range of models, including those with complex and non-conjugate likelihoods. Second, by subsampling data it can be applied to large datasets that might be impractical for traditional methods like MCMC [12, 16].

## 3 Sources of gradient variance in BBVI

Let $\mathbb{V}_{\mathsf{n}, \epsilon}[\nabla f(w; \mathsf{n}, \epsilon)]$ denote the variance[1] of the naive estimator from Eq. 3. The two sources for this variance correspond to data subsampling ($n$) and Monte Carlo noise ($\epsilon$). It is natural to ask how much variance each of these sources contributes. We study this by (numerically) integrating out each of these random variables individually and comparing the variances of the resulting estimators.

Let $f(w; n) = \mathbb{E}_{\epsilon} f(w; n, \epsilon)$ be the objective for a single datum $n$ with Monte Carlo noise integrated out. This can be thought of as an estimator for datum $n$ with a "perfect" control variate. Similarly, let $f(w; \epsilon) = \mathbb{E}_{\mathsf{n}} f(w; \mathsf{n}, \epsilon)$ be the objective for a fixed $\epsilon$ evaluated on the full dataset. In Fig. 1 we generate a single optimization trace using our gradient estimator (described below). Then, for each iteration, we estimate the variance of $\nabla f(w; \mathsf{n}, \epsilon)$, $\nabla f(w; \epsilon)$, and $\nabla f(w; \mathsf{n})$ [2] using a large number of samples. In Table 1 we show the variance at the final iterate on a variety of datasets. (For large datasets, it is too expensive to compute the variance this way at each iteration.)

Our empirical findings suggest that, despite the exact mix of the two sources being task dependent, subsampling noise is usually larger than MC noise. They also show the limits of reducing a single source of noise: No control variate applied to each datum could do better than $\nabla f(w; n)$, while no incremental-gradient-type method could do better than $\nabla f(w; \epsilon)$.

---

[1]For a vector-valued random variable $\mathsf{z}$, we let $\mathbb{V}[\mathsf{z}] = \operatorname{tr} \mathbb{C}[\mathsf{z}]$

[2]Aligned with the experiments in Sec. 7, our evaluation of subsampling variance uses mini-batches, i.e. $\mathbb{V}_{\mathsf{B}} [\mathbb{E}_{\mathsf{n} \in \mathsf{B}} \nabla f(w; \mathsf{n})]$, where $\mathsf{B}$ are mini batches sampled without replacement from $\{1, \ldots, N\}$.

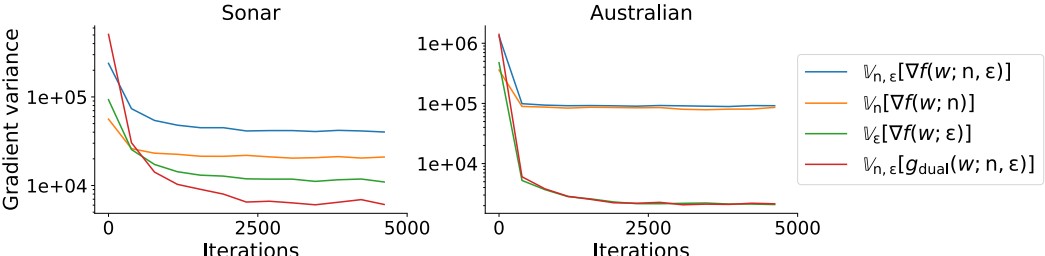

Figure 1: **Gradient Variance Decomposition in Bayesian Logistic Regression using Mean-field BBVI.** The orange line denotes variance from data subsampling ($n$), and the green line denotes Monte Carlo (MC) noise variance ($\epsilon$). For Sonar, both noise sources exhibit similar scales with a batch size of 5. However, for Australian, subsampling noise dominates. Regardless, our proposed gradient estimator $g_{\text{dual}}$ (red line, Eq. (4)) mitigates subsampling noise and controls MC noise, aligning closely with or below the green line (i.e. the variance without data subsampling) in both datasets.

| Task | $\mathbb{V}_{n,\epsilon}[\nabla f(w; n, \epsilon)]$ | $\mathbb{V}_n[\nabla f(w; n)]$ | $\mathbb{V}_\epsilon[\nabla f(w; \epsilon)]$ |
|---|---|---|---|
| Sonar | $4.04 \times 10^4$ | $2.02 \times 10^4$ | $1.16 \times 10^4$ |
| Australian | $9.16 \times 10^4$ | $8.61 \times 10^4$ | $2.07 \times 10^3$ |
| MNIST | $4.21 \times 10^8$ | $3.21 \times 10^8$ | $1.75 \times 10^4$ |
| PPCA | $1.69 \times 10^{10}$ | $1.68 \times 10^{10}$ | $3.73 \times 10^7$ |
| Tennis | $9.96 \times 10^7$ | $9.59 \times 10^7$ | $8.56 \times 10^4$ |

Table 1: BBVI gradient variance decomposition across various tasks, computed at the optimization endpoint. Using a batch size of 100, step size of $1\mathrm{e}{-2}$ for MNIST, PPCA, and Tennis, and a batch size of 5, step size of $5\mathrm{e}{-4}$ for Sonar and Australian. We generally observe subsampling noise $\mathbb{V}_n[\nabla f(w; n)]$ surpassing MC noise $\mathbb{V}_\epsilon[\nabla f(w; \epsilon)]$.

## 4  Dual Control Variate

We now introduce the *dual control variate*, a new approach for controlling the variance of gradient estimators for BBVI. Control variates [24] can reduce the variance of a gradient estimator by adding a zero-mean random variable that is negatively correlated with the gradient estimator. Considering that the objective of BBVI is a function of both $n$ and $\epsilon$, an ideal control variate should also be a function of these variables. We take two steps to construct such a control variate.

1. Inspired by existing control variates for BBVI [19, 9], we create an approximation $\tilde{f}(w; n, \epsilon)$ of the true objective $f(w; n, \epsilon)$, designed so that the expectation $\mathbb{E}_\epsilon \nabla \tilde{f}(w; n, \epsilon)$ can easily be computed for any datum $n$. A common strategy for this is a Taylor-appproximation—to replace $f$ with a low-order polynomial. Then, if the base distribution $s(\epsilon)$ is simple, the expectation $\mathbb{E}_\epsilon[\nabla \tilde{f}(w; n, \epsilon)]$ is often available in closed-form.

2. Inspired by SAGA [6], we maintain a table $W = \{w^1, \dots, w^N\}$ that stores the variational parameters at the last iteration each of the data points $x_1, \cdots, x_N$ were accessed. We also maintain a running average of gradient estimates evaluated at the stored parameters, denoted by $M$. Unlike SAGA, however, this running average is for the gradients of the *approximation $\tilde{f}$*, with the Monte Carlo noise $\epsilon$ integrated out, i.e. $M = \mathbb{E}_n \mathbb{E}_\epsilon \nabla \tilde{f}(w^n; n, \epsilon)$.

Intuitively, as optimization nears the solution, the weights $w$ tend to change slowly. This means that the entries $w^n$ in $W$ will tend to become close to the current iterate $w$. Thus, if $\tilde{f}$ is a good approximation of the true objective, we can expect $\nabla f(w; n, \epsilon)$ to be close to $\nabla \tilde{f}(w^n; n, \epsilon)$, meaning the two will be strongly correlated. However, thanks to the running average $M$, the full expectation of $\nabla \tilde{f}(w^n; n, \epsilon)$ is available in closed-form. This leads to our proposed gradient estimator

$$g_{\text{dual}}(w; n, \epsilon) = \nabla f(w; n, \epsilon) + \underbrace{\mathbb{E}_m \mathbb{E}_\eta \nabla \tilde{f}(w^m; m, \eta) - \nabla \tilde{f}(w^n; n, \epsilon)}_{\text{zero mean control variate } c_{\text{dual}}(w; n, \epsilon)}. \tag{4}$$

---

**Algorithm 1** Black-box variational inference with the dual control variate.

---

**Require:** Learning rate $\lambda$, variational family $q_w(z)$, target $p(z, x)$
**Require:** Estimator $f(w; n, \epsilon)$ whose expectation over $n$ and $\epsilon$ is the negative ELBO from $q_w$ and $p$ (Eq. 2)
**Require:** Approximate estimator $\tilde{f}(w; n, \epsilon)$ that has an expectation over $\epsilon$ in closed form
    Initialize the parameter $w_0$, the parameter table $W = \{w^1, \ldots, w^N\}$
    Initialize running mean $M = \mathbb{E}_m \mathbb{E}_\eta \nabla \tilde{f}(w_0; m, \eta)$     ▷ Closed-form expectation over $\eta$, explicit sum over $m$
    **for** $k = 1, 2, \cdots$ **do**
        Sample $n$ and $\epsilon$
        Extract the value of $w^n$ from the table $W$
        Compute the base gradient $g \leftarrow \nabla f(w_k; n, \epsilon)$
        Compute the control variate $c \leftarrow \mathbb{E}_m \mathbb{E}_\eta \nabla \tilde{f}(w^m; m, \eta) - \nabla \tilde{f}(w^n; n, \epsilon)$ using $\mathbb{E}_m \mathbb{E}_\eta \nabla \tilde{f}(w^m; m, \eta) = M$
        Update the running mean $M \leftarrow M + \frac{1}{N}\left(\mathbb{E}_\eta \nabla \tilde{f}(w_k; n, \eta) - \mathbb{E}_\eta \nabla \tilde{f}(w^n; n, \eta)\right)$     ▷ Closed-form over $\eta$
        Update the table $w^n \leftarrow w_k$
        Update the parameter $w_{k+1} \leftarrow w_k - \lambda(g + c)$.   ▷ Or use $g + c$ in any stochastic optimization algorithm
    **end for**

---

The running average $M = \mathbb{E}_n \mathbb{E}_\epsilon \nabla \tilde{f}(w^n; n, \epsilon)$ can be cheaply maintained through optimization, since a single value $w^n$ changes per iteration and $\mathbb{E}_\epsilon \nabla \tilde{f}(w; n, \epsilon)$ is known in closed form. The variance of the proposed gradient estimator is given by

$$\mathbb{V}[g_{\text{dual}}] = \mathbb{V}_{\epsilon, n}[\nabla f(w; n, \epsilon) - \nabla \tilde{f}(w^n; n, \epsilon)]. \tag{5}$$

Critically, this expression illustrates that the variance of $g_{\text{dual}}$ can be arbitrarily small, only limited by how close $\tilde{f}$ is to $f$ and how close the stored values $w^n$ are to the current parameters $w$.

We illustrate how this gradient estimator can be used for black-box variational inference in Alg. 1. The same idea could be applied more generally to doubly-stochastic objectives in other domains, using the more generic version of the algorithm given in Appendix. D.

## 5 Variance reduction for stochastic optimization

This section considers existing variance reduction techniques and how they compare to the proposed dual estimator.

### 5.1 Monte Carlo sampling and approximation-based control variates

Consider the variational objective from Eq. 2 where we sum over the full dataset in each iteration to define the objective $f(w) = \mathbb{E}_\epsilon f(w; \epsilon)$. The gradient estimator obtained by sampling $\epsilon$ has been observed to sometimes have problematic variance. Previous work [21, 31, 11, 4] proposed to reduce this variance by constructing a (zero-mean) control variate $c(w; \epsilon)$ and defining the new estimator

$$g(w; \epsilon) = \nabla f(w; \epsilon) + c(w; \epsilon). \tag{6}$$

The hope is that $c(w; \epsilon) \approx \nabla f(w) - \nabla f(w; \epsilon)$ approximates the noise of the original estimator, which can lead to large reductions in variance and thus more efficient and reliable inference.

A general way to construct control variates involves using an approximation function $\tilde{f} \approx f$ for which the expectation $\mathbb{E}_\epsilon \tilde{f}(w, \epsilon)$ is available in closed-form [19, 9]. Then, the control variate is defined as $c(w; \epsilon) = \mathbb{E}_\eta \nabla \tilde{f}(w; \eta) - \nabla \tilde{f}(w; \epsilon)$, and the estimator from Eq. (6) becomes

$$g(w; \epsilon) = \nabla f(w; \epsilon) + \mathbb{E}_\eta \nabla \tilde{f}(w; \eta) - \nabla \tilde{f}(w; \epsilon). \tag{7}$$

Intuitively, the better $\tilde{f}$ approximates $f$, the lower the variance of this estimator tends to be (for a perfect approximation, the variance is fully removed). A popular choice for $\tilde{f}$ involves a quadratic function, either learned [9] or obtained through a second order Taylor expansion [19], since their expectation under general Gaussian variational distributions is tractable.

In doubly-stochastic problems with objectives of the form $f(w; n, \epsilon)$, data $n$ is subsampled as well as $\epsilon$. While the above control variate has most commonly been used without subsampling, it can also be used with subsampling, by developing an approximation $\tilde{f}(w; n, \epsilon)$ to $f(w; n, \epsilon)$ for each datum $n$. This leads to the control variate $\mathbb{E}_\eta \nabla \tilde{f}(w; n, \eta) - \nabla \tilde{f}(w; n, \epsilon)$ and gradient estimator

$$g_{\text{cv}}(w; n, \epsilon) = \nabla f(w; n, \epsilon) + \underbrace{\mathbb{E}_\eta \nabla \tilde{f}(w; n, \eta) - \nabla \tilde{f}(w; n, \epsilon)}_{\text{zero mean control variate } c_{\text{cv}}(w; n, \epsilon)}. \tag{8}$$

It is important to note that this control variate is unable to reduce variance coming from data subsampling. Even if $\tilde{f}(w; n, \epsilon)$ were a *perfect* approximation there would still be gradient variance due to $n$ being sampled randomly. This can be shown by noting that the variance of this estimator is given by (see Appendix. B.1 for a full derivation using the law of total variance)

$$\mathbb{V}[g_{\text{cv}}] = \mathbb{E}_n \mathbb{V}_\epsilon[\nabla f(w; n, \epsilon) - \nabla \tilde{f}(w; \mathsf{n}, \epsilon)] + \mathbb{V}_n[\nabla f(w; \mathsf{n})] \geq \mathbb{V}_n[\nabla f(w; \mathsf{n})]. \tag{9}$$

While the first term of the expression above can be made arbitrarily small in the ideal case of a perfect approximation $\tilde{f} \approx f$, the second term is irreducible, regardless of the quality of the approximation used. Therefore, this approach cannot reduce subsampling variance. As shown in Fig. 2 and Table 1, subsampling variance is typically substantial, and often several orders of magnitude larger than Monte-Carlo variance. When this is true, this control variate, which is only able to reduce variance coming from Monte Carlo sampling, will have minimal effect on the overall gradient variance.

## 5.2 Data subsampling and incremental gradient methods

We now consider a stochastic optimization problem with objective $f(w) = \mathbb{E}_\mathsf{n} f(w; \mathsf{n})$, where $\mathsf{n}$ is uniformly distributed on $\{1, \ldots, N\}$, representing data indices, but no other stochasticity (i.e. no Monte Carlo sampling). While one could compute $f$ or its gradient exactly, this is expensive when $N$ is large. A popular alternative involves drawing a random $\mathsf{n}$ and using the estimator $\nabla f(w; \mathsf{n})$ with a stochastic optimization method, such as stochastic gradient descent. Alternatively, for such problems, *incremental gradient* methods [26, 28, 13, 7, 10] often lead to faster convergence.

While details vary by algorithm, the basic idea of incremental gradient methods is to "recycle" previous gradient evaluations to reduce randomness. For example, SAGA [6] stores the parameters $w^n$ of the most recent iteration where $f(w; n)$ was evaluated and takes a step as

$$w \leftarrow w - \lambda \left( \nabla f(w; n) + \mathbb{E}_\mathsf{m} \nabla f(w^\mathsf{m}; \mathsf{m}) - \nabla f(w^n; n) \right), \tag{10}$$

where $\lambda$ is a step size and the expectation over $m$ is tracked efficiently using a running average, meaning the cost per iteration is independent of $N$. The update rule above can be interpreted as using a control variate to reduce the variance of the naive estimator $\nabla f(w; n)$ as

$$g(w; n) = \nabla f(w; n) + \underbrace{\mathbb{E}_\mathsf{m} \nabla f(w^\mathsf{m}; \mathsf{m}) - \nabla f(w^n; n)}_{\text{zero mean control variate}}. \tag{11}$$

When $w^m \approx w$, the first and last terms in Eq. (11) will approximately cancel, leading to a gradient estimator with significantly lower variance.

We now consider a doubly-stochastic objective $f(w; n, \epsilon)$. In principle, one might consider computing the estimator from Eq. (11) for each value of $\epsilon$, i.e. using the gradient estimator

$$g_{\text{inc}}(w; n, \epsilon) = \nabla f_n(w; n, \epsilon) + \underbrace{\mathbb{E}_\mathsf{m} \nabla f(w^\mathsf{m}; \mathsf{m}, \epsilon) - \nabla f(w^n; n, \epsilon)}_{\text{zero mean control variate } c_{\text{inc}}(w; n, \epsilon)}. \tag{12}$$

This has two issues. First, the resulting method does not address Monte Carlo noise due to sampling $\epsilon$. This can be shown by noting that the variance of this estimator is given by (see Appendix B.2)

$$\mathbb{V}[g_{\text{inc}}] = \mathbb{E}_\epsilon \mathbb{V}_\mathsf{n}[\nabla f(w; \mathsf{n}, \epsilon) - \nabla f(w^\mathsf{n}; \mathsf{n}, \epsilon)] + \mathbb{V}_\epsilon[\nabla f(w; \epsilon)] \geq \mathbb{V}_\epsilon[\nabla f(w; \epsilon)]. \tag{13}$$

Since the second term in the variance expression above is irreducible, the variance cannot be expected to go to zero, no matter how close all the stored vectors $w^n$ are to the current parameters. Intuitively, this approach cannot do better than simply evaluating the objective on the full dataset for a random $\epsilon$.

158 The second issue is more critical: $g_{\mathrm{inc}}$ *cannot be implemented efficiently*. The value of $\nabla f(w^{\mathsf{n}}; n, \epsilon)$
159 is dependent on $\epsilon$, which is resampled at each iteration. Therefore, it is not possible to efficiently
160 maintain $\mathbb{E}_{\mathsf{m}} \nabla f(w^{\mathsf{m}}; \mathsf{m}, \epsilon)$ needed by Eq. (12) as a running average. In general, this can only
161 be computed by looping over the full dataset in each iteration. While possible, this destroys the
162 computational advantage of subsampling. For some models with special structure [32, 34] it is
163 possible to efficiently maintain the needed running gradient. However, this can only be done in
164 special cases with model-specific derivations, breaking the universality of BBVI.

165 It may seem odd that $g_{\mathrm{inc}}$ has these computational issues, while $g_{\mathrm{dual}}$—an estimator intended to
166 reduce variance even further—does not. The fundamental reason is that the dual estimator only stores
167 (approximate) gradients after integrating over the Monte Carlo variable $\epsilon$, so the needed running
168 average is independent of $\epsilon$.

### 5.3 Ensembles of control variate

170 It is possible to combine multiple control variates. For example, [8] combined control variates that
171 reduced Monte Carlo noise [19] with one that reduced subsampling noise [32] (for a special case
172 where $g_{\mathrm{inc}}$ is tractable). While this approach can be better than either control variate alone, it still does
173 not reduce *joint* variance. To see this, consider a gradient estimator that uses a convex combination of
174 the two above control variates. For any $\beta \in (0, 1)$ write

$$g_{\mathrm{combo}}(w; n, \epsilon) = \nabla f(w; n, \epsilon) + \underbrace{\beta c_{\mathrm{cv}}(w; n, \epsilon) + (1 - \beta) c_{\mathrm{inc}}(w; n, \epsilon)}_{c_{\mathrm{combo}}(w; n, \epsilon)}. \tag{14}$$

175 It can be shown (Appendix B.3) that if both $c_{\mathrm{cv}}$ and $c_{\mathrm{inc}}$ are "perfect", that is, if $\tilde{f}(w; n, \epsilon) = f(w; n, \epsilon)$ and $w^n = w$ for all $n$, then
176

$$\mathbb{V}[g_{\mathrm{combo}}] = \beta^2 \mathbb{V}_{\mathsf{n}}[\nabla f(w; \mathsf{n})] + (1 - \beta)^2 \mathbb{V}_{\epsilon}[\nabla f(w; \epsilon)]. \tag{15}$$

177 Even in this idealized scenario, such an estimator cannot reduce variance to zero, because each of
178 the individual control variates leaves one source of noise uncontrolled. The dual control variate
179 overcomes this because it models interactions between $\epsilon$ and $n$.

## 6   Related work

181 Recent work proposed to approximate the optimal batch-dependent control variate for BBVI using
182 a recognition network [4]. Similar to our work, they take into account the usage of subsampling
183 when designing their variance reduction techniques for BBVI. However, like $g_{\mathrm{cv}}$, their control variate
184 reduces the *conditional* variance of MC noise (conditioned on $n$) but is unable to reduce subsampling
185 noise (like $g_{\mathrm{cv}}$).

186 It is also worth discussing a special incremental gradient method called SMISO [1], designed
187 for doubly-stochastic problems. Intuitively, SMISO uses exponential averaging to approximately
188 marginalize out $\epsilon$, and then runs MISO/Finito [7, 18] (an incremental gradient method similar to
189 SAGA) to control the subsampling noise. While the method is similar to running SGD with an
190 incremental control variate, it is not obvious how to separate the control variate from the algorithm,
191 meaning we cannot use the SMISO idea as a control variate to get a gradient estimator that can be
192 used with other optimizers like Adam, we include a detailed discussion on this issue in Appendix. A.
193 Nevertheless, we still include SMISO as one of our baselines.

## 7   Experiments

195 This section empirically demonstrates the effectiveness of the dual control variate for BBVI. We
196 focus on mean-field Gaussian BBVI, where the variational posterior follows a multivariate Gaussian
197 with diagonal covariance $q_w(z) = \mathcal{N}(\boldsymbol{\mu}, \mathrm{diag}(\boldsymbol{\sigma}^2))$, with parameters $w = (\boldsymbol{\mu}, \log(\boldsymbol{\sigma}))$.

198 The gradient estimators $g_{\mathrm{cv}}(w; n, \epsilon)$ and $g_{\mathrm{dual}}(w; n, \epsilon)$ require an approximation function with
199 expectation over $\epsilon$ available in closed form. Inspired by previous work [19], we get an approximation
200 for $f(w; n, \epsilon)$ using a second order Taylor expansion for the negative total likelihood $k_n(z) =$

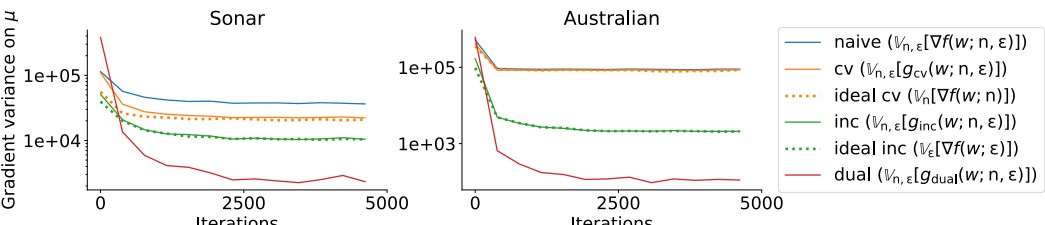

Figure 2: **Dual control variate helps reduce gradient variance.** The naive gradient estimator (Eq. (3)) is the baseline, while the cv estimator (Eq. (8)) controls the Monte Carlo noise, the inc estimator (Eq. (12)) controls for subsampling noise, and the proposed dual estimator (Eq. (4)) controls for both. The variance of cv and inc, as is shown in Eq. (9) and Eq. (13) are lower-bounded by the dotted lines, while dual is capable of reducing the variance to significantly lower values, leading to better and faster convergence (Fig. 3).

$N \log p(x_n \mid z) + \log p(z)$ around $z_0 = \mathcal{T}_w(0)$[3], which yields

$$\tilde{f}(w; n, \epsilon) = k_n(z_0) + (\mathcal{T}_w(\epsilon) - z_0)^\top \nabla k_n(z_0) + \frac{1}{2}(\mathcal{T}_w(\epsilon) - z_0)^\top \nabla^2 k_n(z_0)(\mathcal{T}_w(\epsilon) - z_0)^\top + \mathbb{H}(w), \ (16)$$

where we assume the entropy can be computed in closed-form. For a mean-field Gaussian variational distribution, the expected gradient of the approximation Eq. (16) can only be computed efficiently (via Hessian-vector products) with respect to the mean parameter $\boldsymbol{\mu}$ but not for the scale parameter $\boldsymbol{\sigma}$, which means $g_{\mathrm{cv}}(w; n, \epsilon)$ and $g_{\mathrm{dual}}(w; n, \epsilon)$ can only be used as the gradient estimator for $\boldsymbol{\mu}$. Fortunately, controlling only the gradient variance on $\boldsymbol{\mu}$ often means controlling most of the variance, as, with mean-field Gaussians, the total gradient variance is often dominated by variance from $\boldsymbol{\mu}$ [9].

## 7.1 Experiment setup

We evaluate our methods by performing BBVI on a range of tasks: binary Bayesian logistic regression on two datasets, Sonar (number of samples $N = 208$, dimensionality $D = 60$) and Australian ($N = 690, D = 14$); multi-class Bayesian logistic regression on MNIST [17] ($N = 60000, D = 7840$); probabilistic principal component analysis [29] (PPCA, $N = 60000, D = 12544$); and Bradley-Terry model [5] for tennis player ranking (Tennis, $N = 169405, D = 5525$). We give full model descriptions in Sec. 7.3.

**Baselines.** We compare $g_{\mathrm{dual}}$ (Eq. (4)) with $g_{\mathrm{naive}}$ (Eq. (3)) and $g_{\mathrm{cv}}$ (Eq. (8)). For Sonar and Australian (small datasets) we include $g_{\mathrm{inc}}$ (Eq. (12)) as well, which requires a full pass through the full dataset at each iteration. For larger-scale problems, $g_{\mathrm{inc}}$ becomes intractable, so we use SMISO instead.

**Optimization details.** We optimize using Adam [14] for the larger-scale MNIST, PPCA, and Tennis datasets and SGD without momentum for the small-scale Sonar and Australian dataset for transparency. The optimizer for SMISO is pre-determined by its algorithmic structure and cannot be changed. For all estimators, we perform a step-size search (see Appendix C) to ensure a fair comparison and use a single shared $\epsilon$ for all samples in the batch.

**Mini-batching.** In practice, for efficient implementation on GPUs, we draw a mini-batch B of data at each iteration (reshuffling for each epoch). For inc, dual, and SMISO, we update multiple entities in the parameter table per iteration and adjust the running mean accordingly. For the Sonar and Australian datasets, due to their small sizes, we use $|\mathsf{B}| = 5$. For other datasets we use $|\mathsf{B}| = 100$.

**Evaluation metrics.** We track the ELBO on the full dataset, explicitly computing $\mathbb{E}_n$ (summing over the full dataset) and approximating $\mathbb{E}_\epsilon$ with 5000 Monte Carlo samples. We present ELBO vs. iterations plots for a single example learning rate as well as ELBO values for the best learning rate chosen retrospectively for each iteration. In addition, we present the final ELBO after training vs. step size at different iterations. For the Sonar and Australian datasets, given the small size, we include a detailed trace of gradient variance on $\boldsymbol{\mu}$ across different estimators. This enables empirical validation of the lower bounds derived in Eq. (9) and Eq. (13).

---

[3]We use $z_0 = \mathrm{stop\_gradient}(\mathcal{T}_w(0))$ so that the gradient does not backpropagate from $z_0$ to $w$.

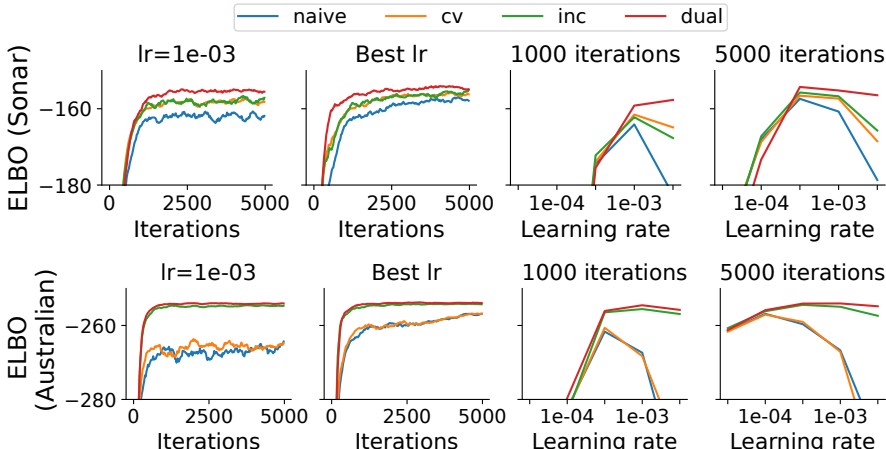

Figure 3: **With reduced variance (Fig. 2), the dual estimator provides better convergence at a larger step size.** On Sonar, Monte Carlo noise and subsampling noise are of similar scale, therefore jointly controlling them shows better performance than methods that only control one source of noise. On Australian, where the subsampling noise dominates, dual shows similar performance compared with inc, which controls subsampling noise but *cannot* be efficiently computed (requires pass over the full dataset at each iteration).

**Initialization.** The variational parameters are randomly initialized using a standard Gaussian and all results reported are averages over multiple independent trials: We run 10 trials for Sonar and Australian, and 5 for the larger scale problems due to resource constraint.

## 7.2 Results

The experiment results for Sonar and Australian are presented in Fig. 2 and Fig. 3. Both the inc and cv estimators have lower variance than the naive estimator, but the improvement varies by the dataset. The excellent performance of the (impractical) inc estimator on Australian shows the importance of reducing subsampling noise. Overall, the dual estimator has the lowest variance, which enables larger learning rates and thus faster optimization.

Similar results can be observed on MNIST, PPCA, and Tennis in Fig. 4 (for these datasets inc is intractable, so we include SMISO as a baseline instead). Again, dual yields faster and better convergence than naive and cv. Whereas SMISO, which does not adopt momentum nor adaptive step size, suffers from slow convergence speed in that it has to utilize a small step size to prevent diverging during optimization. We provide comparisons of different estimators using SGD in Appendix. E.

## 7.3 Model descriptions

**Binary/Multi-class Bayesian logistic regression.** A standard logistic regression model with standard Gaussian prior.

**Probabilistic principal component analysis (PPCA).** Given a centered dataset $\boldsymbol{x}_1, \ldots, \boldsymbol{x}_N \in \mathbb{R}^D$, PPCA [29] seeks to extract its principal axes $\boldsymbol{W} \in \mathbb{R}^{D \times K}$ by assuming $\mathbf{x}_n \sim \mathcal{N}(\mathbf{0}, \boldsymbol{W}\boldsymbol{W}^\top + \text{diag}(\lambda^2))$. In our experiments, we employ a standard Gaussian prior on $\boldsymbol{W}$ and use BBVI to approximate the posterior over $\boldsymbol{W}$. We then test PPCA on the standardized training set of MNIST with $K = 16$ and $\lambda = 1$.

**Bradley Terry model (Tennis).** This is a model used to rank players from pair-wise matches. Each player is represented by a score $\theta_i$, and each score is assigned a standard Gaussian prior. The result of a match between two players is modeled by the inverse logit of their score difference $\mathsf{y}_n \sim \text{Bernoulli}(\text{logit}^{-1}(\theta_i - \theta_j))$ where $\mathsf{y}_n = 1$ denotes a win by player $n$. We subsample over matches and perform inference over the score of each player. We evaluate the model on men's tennis matches log starting from 1960, which contains the results of $169405$ matches among $5525$ players.

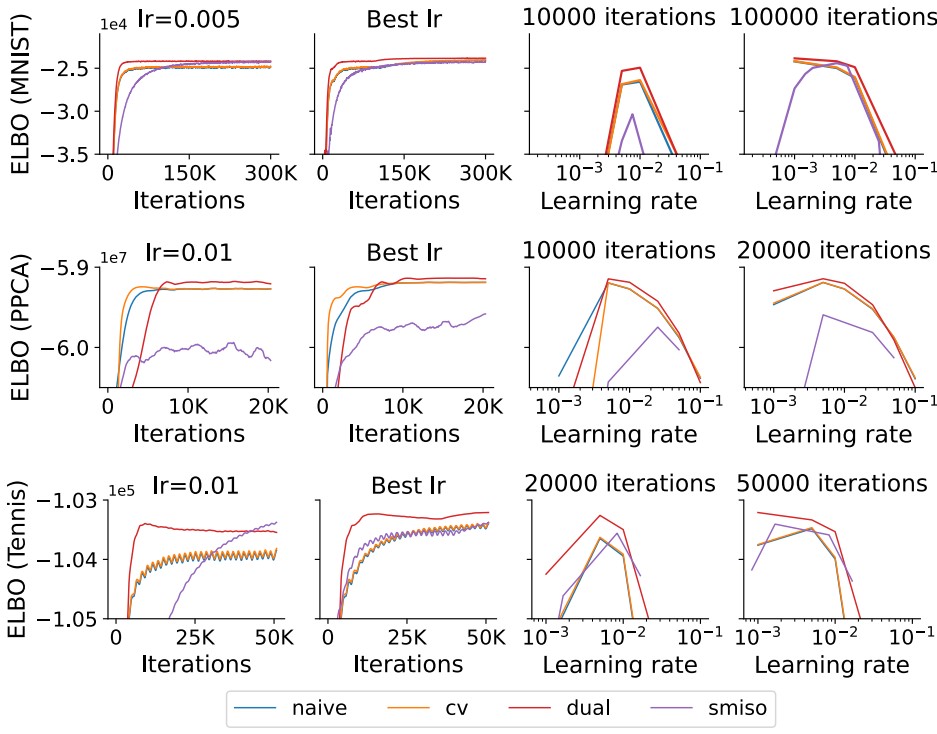

Figure 4: **On larger scale problems, the dual estimator leads to improved convergence.** In large-scale problems, cv shows little or no improvement upon naive while dual converges faster. We suspect that most of the improvement in the dual estimator comes from reducing subsampling variance. SMISO shows slow convergence. We suspect that is because it is an "SGD-type" algorithm while all others use Adam. Note that the step size for SMISO is rescaled for visualization. The loss shows periodic structure in Tennis, this happens because gradients have correlated noise that cancels out at the end of each epoch.

| Estimator | Variance lower bound | $\nabla f$ evals per iteration | Wall-clock time per iteration | | |
|---|---|---|---|---|---|
| | | | MNIST | PPCA | Tennis |
| naive | $\mathbb{V}_{n,\epsilon}[\nabla f(w; n, \epsilon)]$ | 1 | 10.4ms | 12.8ms | 10.2ms |
| cv | $\mathbb{V}_n[\nabla f(w; n)]$ | 2 | 12.8ms | 18.5ms | 14.6ms |
| inc | $\mathbb{V}_\epsilon[\nabla f(w; \epsilon)]$ | N+2 | 328ms | 897ms | 588ms |
| dual | 0 | 3 | 17.6ms | 31.2ms | 29.6ms |
| Fullbatch-naive | $\mathbb{V}_\epsilon[\nabla f(w; \epsilon)]$ | N | 201ms | 740ms | 203ms |
| Fullbatch-$c_{cv}$ | 0 | 2N | 360ms | 1606ms | 246ms |

Table 2: Variance, oracle complexity, and wall-clock time for different estimators. Notice that inc is more expensive than Fullbatch-naive. We hypothesize this is because inc uses separate $w^n$ for different data points, which is less efficient for parallelism.

## 7.4 Efficiency analysis

We now study the computational cost of different estimators. In terms of the number of "oracle" evaluations (i.e. evaluations of $f(w; n, \epsilon)$ or its gradient), the naive estimator is the most efficient, requiring a single oracle evaluation per iteration. The cv estimator requires one gradient and also one Hessian-vector product, while the dual estimator needs one gradient and two Hessian-vector products, one for the control variate and one for updating the running mean $M$.

Additionally, Table 2 shows measured runtimes based on a JAX implementation on an Nvidia 2080ti GPU. All numbers are for a single optimization step, averaged over 200 steps. Overall, each iteration with the dual estimator is between 1.5 to 2.5 times slower than naive, and around 1.2 times slower than cv. Lastly, given that dual achieves a given performance in an order of magnitude fewer iterations (Figs. 3 and 4), it is the fastest in terms of wall-clock time. The exact wall-clock time v.s. ELBO results are presented in Appendix. F.

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
