# OpenReview forum: "Dual control variate for faster black-box variational inference"
_NeurIPS.cc/2023/Conference — Submitted to NeurIPS 2023_

### Official Review · Reviewer_NBCj · 2023-06-21

**Soundness:** 4 excellent
**Presentation:** 3 good
**Contribution:** 3 good
**Rating:** 7
**Confidence:** 5

**Summary:**

The authors introduce a "dual" control variate for reducing gradient variance in black-box variational inference in the context of models that admit data subsampling (i.e. exhibit the required conditional independence). The dual control variant is joint in that it simultaneously addresses the two sources of Monte Carlo variance in ELBO approximations: that due to latent variable sampling and that due to data subsampling. The basic idea, in essence, is to leverage linear or quadratic ELBO approximations (which admit closed form evaluation for e.g. gaussian mean field variational families) in conjunction with a running average of gradient estimates for each data point using the variational parameters from the last iteration in which each data point was accessed. Experiments demonstrate that the proposed method can substantially reduce gradient variance (in particular that due to data subsampling, which is often dominant), thus yielding better ELBO optimization (both w.r.t. wallclock time and final ELBOs obtained).

**Strengths:**

The main strengths of this submission include the following:
- it addresses a general problem of relatively wide interest in the NeurIPS community (namely, how best to do black-box variational inference)
- it addresses a particular component of that problem that is often somewhat overlooked compared to other aspects (namely how best to do optimization)
- the experiments are pretty convincing in establishing the efficacy of the method
- the suggested method is technically sound and would appear to be pretty simple to implement
- the discussion in Sec 5 and the variance analysis (appendix B) help the reader conceptually place the proposed control variate alongside other alternatives

**Weaknesses:**

The main weaknesses of this submission, as I see them, are the following:
- the notation is a bit confusing in some places
- some of the limitations of the method and/or extensions to more general problem settings are either not discussed or are insufficiently discussed

Let me expand on these points:

While the notation for this kind of paper will necessarily be somewhat clunky given the various sources of sampling variability that have to be carefully tracked, I think some improvements are possible. In particular I find the choice of M for the running gradient particularly suboptimal. Since capital N is a positive integer and little n and little m are used to index integers, one might expect that M is also a positive integer. I suggest that M be replaced by something like G(bold $w$) to avoid this confusion and to emphasize that G depends on the "table" of $w$s.

The authors consider a generic but still somewhat limited problem specification, in particular they do not consider local latent variables, model learning, or amortization. More discussion of these points would probably be of considerable interest to the reader (for more discussion see below). In addition one of the weaknesses of this method is the potentially large memory requirements, which are O(Nd) where d is the latent dimension. This needs to be *very clearly emphasized*.



**Questions:**

- Do you expect your method would work more or less unchanged for hierarchical models that feature both global and local latent variables? Or would the algorithm need to be adapted to that setting?
- Arguably besides the possibility of data subsampling for appropriate model classes, some of the biggest selling points of variational inference are amortization and model learning (i.e. learning point estimates for model parameters).
  - I guess you could support model learning directly without any change to your algorithm, although I suspect the performance of your control variate would tend to decrease since now in addition to "$w$ drift" you'd also have "$\theta$ drift" (where theta is the model parameter). Can you please comment?
  - I guess you could also support amortization (e.g. of local latent variables in hierarchical models) without any changes to your algorithm, although since amortization typically involves a neural network with a largeish number of parameters, this would make the memory requirements needed to cheaply compute the running average M intractably large. Can you please comment?
- typo: appproximation
- It would be great to see some ablation studies comparing first order to second order taylor approximations. How important is going to second order in practice?
- What kind of runtime performance would you expect if used a multivariatiate gaussian distribution for the variational distribution? Would the dual estimator still be ~1.5 to ~2.5 times slower than computing control variant free gradients? Or would the gap increase?
- It should be possible to do a more fine-grained theoretical analysis for a model that admits more analytic control, e.g. bayesian linear regression. Such an analysis could be particularly valuable in further delineating the regimes in which the dual variate is expected to perform best.
- You state that "the total gradient variance is often dominated by variance from μ" [line 207]. Do you have any intuition for this claim?
- Comment: For a large dataset initializing the running mean M at the beginning seems like it could be a waste of compute time. It might make more sense to not use the control variate during the first epoch and use the first epoch to collect M.

**Limitations:**

As discussed in the weaknesses section, I believe some of the limitations of the method (e.g. w.r.t.~memory requirements and likely trouble with amortization) should be discussed and/or better emphasized.

---

> ### Author Rebuttal · Authors · 2023-08-08
>
> Thank you for your careful review and suggestions.
>
> - Usage of "M": This notation comes from the SAGA paper, but we agree it likely causes confusion here. We plan to replace it with $g(w)$ or $\bar{g}$ in later revisions.
>
> - Local latent variables: We agree that this is a limitation. In our PPCA experiments that involve local latent variables, we tackle the local latent $z_i$ by analytically marginalizing it out. However, subsampling for local latent variables is generally tricky and would require additional techniques such as amortization (see citations (1,2) below). We suspect this might be addressed by combining this dual control variate with that kind of amortization strategy. We will add a discussion on this point to the discussion section.
>
> - Limitation on model learning: Our method could be applied with minimal changes in a model learning setting where one wishes to optimize the parameters of p at the same time as learning q. Roughly speaking, let p have parameters θ, and store and update those similarly to how w is updated now. However, it is possible that the method would tend to be less effective in that setting, since changes in p will mean that the stored parameters will tend to get "out of date" somewhat faster. So we don't intend to make any strong claims about practical performance.
>
> - Memory usage: We agree that SAGA takes O(Nd) memory and will emphasize this in later revisions. However, note we can also use an SVRG-like strategy which does not require additional memory. We provide a pseudo-code for SVRG version of dual control variate (Algorithm 2) below and provide experiment results with it on Australian (Figure. 2) in the rebuttal PDF. The SVRG version shows performance comparable to SAGA, though it has an extra hyperparameter, the outer-loop size, and requires additional gradient evaluations at each iteration.
>
> - First-order v.s. Second-order Taylor approximation: If we use a first-order Taylor as our approximation, the gradient with respect to $\mu$ would be constant, leading to a *constant* control variate of 0, which is identical to not using CV at all.
>
> - Runtime performance for multivariate Gaussian: We believe that the time complexity will be the same and we expect dual to remain the same order of complexity.
>
> - Gradient noise dominated by $\mu$. We do not have satisfying intuition, but we do confirm previous experimental observations. We interpret this as a combination of the fact that with mean-field there are fewer scale parameters and they tend to be smaller in magnitude.
>
> - Initialize using first epoch: Thanks for pointing this out. We are already using this regime where we use naive for the first epoch and use the first epoch to initialize M. We apologize for missing the details, we will include this implementation detail to the algorithm in later revisions.
>
>
> (1) Abhinav Agrawal and Justin Domke. "Amortized variational inference for simple hierarchical models." Advances in Neural Information Processing Systems 34 (2021): 21388-21399.
>
> (2) Charles Margossian and David Blei. "Amortized Variational Inference: When and Why?" arXiv:2307.11018
>
> --------------
>
> ```python
> Algorithm 2: Black-box variational inference with the dual control variate (SVRG version)
>
> Require: Learning rate λ, variational family qw(z), target p(z, x), update frequency m
> Require: Estimator f(w; n, epsilon) whose expectation over n and epsilon is the negative ELBO from qw and p
> Require: Approximate estimator tilde_f(w; n, epsilon) that has an expectation over epsilon in closed form
>
> Initialize the parameter tilde_w
> for s = 1, 2, . . . do
>     Let tilde_w ← tilde_w_{s−1} and w_0 ← tilde_w
>     Compute the full gradient of tilde_f at tilde_w: M = E_m E_eta grad(tilde_f(tilde_w; m, eta))
>     for k = 1, 2, · · · , m do
>         Sample n and epsilon
>         Compute the base gradient g ← grad(f(w_{k−1}; n, epsilon)
>         Compute the control variate c ← M − grad(tilde_f(tilde_w; n, epsilon))
>         Update the parameter w_k ← w_{k−1} − λ (g + c). . Or use g + c in any stochastic optimization algorithm
>     end for
>     Update tilde_w_s ← w_m
> end for
> ```

---

> > ### Comment · Reviewer_NBCj · 2023-08-11
> >
> > Thanks for delineating what a SVRG version would look like.
> >
> > Please do not forgot to discuss memory usage in the revised paper.
> >
> > While the suggested algorithm is not shockingly original, I believe it makes a strong contribution to the literature, would be of interest to many in the NeurIPS community, and could be useful in practice. As such I continue to recommend publication.

---

> > > ### Author Response · Authors · 2023-08-16
> > >
> > > Thank you for the response! We agree that memory is a crucial issue and will most certainly make the O(ND) requirements for the SAGA version unambiguously clear in the later revision.

---

### Official Review · Reviewer_SSK5 · 2023-07-02

**Soundness:** 3 good
**Presentation:** 2 fair
**Contribution:** 3 good
**Rating:** 7
**Confidence:** 4

**Summary:**

This paper proposes a new control variate for black-box variational inference. In particular, the proposed "dual" control variate attempts to reduce the subsampling noise and Monte Carlo noise at the same time. For this, the paper utilizes an incremental gradient-like scheme. The performance of the new control variate is empirically verified on Bayesian inference tasks with large datasets.

**Strengths:**

* While control variates have been an active area of research for BBVI, reducing the subsampling noise has certainly been a problem that hasn't been addressed. In fact, the paper shows that conventional control variate solutions do not solve this problem *at all*, despite the fact that subsampling is a major source of variance.
* The paper motivates the latter point by empirically computing the subsampling noise. Overall, the paper conveys the motivations for the proposed control variate very clearly.
* The proposed control variate based on incremental gradients seems fairly simple to implement, but not trivial. Thus, the proposed strategy has clear technical contribution.
* Empirical evaluations are thorough and adequate to show the superiority of the proposed control variate.

**Weaknesses:**

* Given that the paper builds on top of incremental gradient methods, which, as typical of the optimization community, comes with a heap of theoretical tools for rigorous analysis. Given this fact, it would have been amazing if the paper also provided some rigorous analysis of the proposed strategy.
* There seems to be some room for improvement in terms of paper organization and notation. The notation can be quite confusing at times, for example, lots of things happen behind innocent-looking superscripts. More on this below.

**Questions:**

### Major Comments
* What was the motivation behind calling the control variate "dual"? At least to people who have some experience in optimization, I believe the word dual would automatically trigger the notion of Lagrangian duality. How about "doubly" in accordance with the term "doubly stochastic?"
* The section organization seems to have room for improvement. For example, a general explanation of "general" control variates appears way back in Section 5.1. Considering readers that are not familiar with control variates, it would be better to start with a high-level explanation of how a control variate is supposed to work. A good place would be after Section 3, where the variance of BBVI is motivated to be a problem, and before Section 4 where a control variate is first introduced.
* Similarly, I felt that the whole discussion in Section 5.1 and 5.2 would have been a very good way to inspire the readers before actually unrevealing the dual control variate. While reading Section 5.1 and 5.2 I had to constantly go back to Section 4 to remind how the dual control variate attempts to solve the problem.
* Algorithm 1 is not very helpful to precisely understand the implementation. For example, in "Compute the control variate" what "using ... $=M$ " mean? Does this mean we simply plug $M$ in place of $ E_m E_{\eta} \nabla \tilde{f} \left(w^m; m, \eta\right) $ ? Then why denote  $ E_m E_{\eta} \nabla \tilde{f} \left(w^m; m, \eta\right) $ ? I think here the authors need to clearly distinguish between values and computed quantities. Similarly, the notation for the running mean $M$ is too uninformative. I think something like $\tilde{g}$ or $\hat{g}$ would be more appropriate. This is also useful to understand the dimensionality/domain of each variable.
* Line 207: Doesn't this contradict the conclusions of [2,3]? From my understanding of [2], the $\mathcal{O}\left(d\right)$ dimensional dependence comes from the gradients with respect to the scale.
* Figure 4 Tennis: Do you have any insight as why the ELBO for the dual control variate peaks and then decreases? This seems quite weird given that the control variate is unbiased.
* Caption of Figure 4: This is perhaps a minor point, but could you elaborate why gradient noise of Tennis is correlated? Is this because reshuffling was used here instead of iid subsampling?
* Section 6: Please include a discussion of other non-dual control variates since this works, in the broader scheme of things, is a control variate paper. Also, why not include Section 5.3 in Section 6?

### Minor Comments
* Eq. (1) I was a little bit confused at first because I am so used to see the likelihood adjustment ratio to be N/B, where B is the batch size. I think it would be useful to either use N/B overall so that the notion that we're dealing with batches is more clear, or to mention in the text that the batch size is assumed to be 1.
* Line 59: The classic paper that first proposed doubly stochstic BBVI was [1]. I recommend adding it here.
* Table 1: How about using a bar plot instead of a table? I think it would be more appropriate since the relative magnitude is key here.
* Line 118: I think it would be better to explicitly show that the performance of a control variate is entirely driven by the covariance between the estimate and the control variate.

### References
* [1] Titsias, Michalis, and Miguel Lázaro-Gredilla. "Doubly stochastic variational Bayes for non-conjugate inference." In International conference on machine learning, pp. 1971-1979. PMLR, 2014.
* [2] Domke, Justin. "Provable gradient variance guarantees for black-box variational inference." Advances in Neural Information Processing Systems 32 (2019).
* [3] Bhatia, Kush, Nikki Lijing Kuang, Yi-An Ma, and Yixin Wang. "Statistical and computational trade-offs in variational inference: A case study in inferential model selection." arXiv preprint arXiv:2207.11208 (2022).

**Limitations:**

Yes.

---

> ### Author Rebuttal · Authors · 2023-08-08
>
> Thank you for your positive review and detailed suggestions. Please see below for our response.
>
> - Naming: We agree that the connotation of Lagrangian duality is an unfortunate clash of terminology and are certainly open to changing the name. Our reason for not using "doubly control variate" is that it is very similar to the name "double control variate" which is already used in a different context of discrete latent space models (citation (1) below). We would appreciate further advice on this point.
>
> - Paper organization: Thanks for your suggestion. We agree that introducing details behind control variates earlier in Section 3 may improve clarity. We also believe that moving Sections 5.1 and 5.2 to an earlier part of the paper could help readers better understand the context. We plan to reorganize the manuscript in accordance with this suggestion.
>
> - Notation in Algorithm 1: We agree that M may be unclear, and does not emphasize that M "lives in the same space" as the gradient estimates. Our choice of M followed the original SAGA paper, but we accept that a change would be better here. We plan to change the notation to  $g(\mathbf{w})$ or $\bar{g}$ following reviewer NBCj's suggestion.
>
> - Gradient variance from scale: It does not contradict [2, 3]. Those papers refer to full-rank Gaussians, whereas the statement in line 207 refers to the mean-field case as is discussed in "Approximation Based Variance Reduction for Reparameterization Gradients".
>
> - ELBO peaks and decreases in Fig. 4, Tennis, left panel. We believe this happens because the step size 0.01 is too large. This behavior disappears in the second plot when we choose the best step size retrospectively. We also suspect that this may be caused by the use of Adam as the optimizer—as gradient variance decreases over time (especially with our control variate), the effective step size increases, meaning the size of the random fluctuations increases even with a fixed nominal step size.
>
> - Correlated gradient noise in Tennis: We also found this behavior very strange at first. Yes, this is due to using dataset reshuffling instead of i.i.d sampling for implementing mini-batching. Essentially what happens is that midway through the epoch only a random subset of matches for each player has been sampled, which pushes the iterates a bit away from the solution, but at the end of each epoch, all matches for each player have been sampled and so that noise gets canceled out. (The cyclic behavior is exactly one epoch long.) Incidentally, note that this periodic behavior is almost completely eliminated by the dual control variate since it controls for the noise introduced by this kind of randomness in the minibatch selection. (It is somewhat reduced by S-miso as well.)
>
> - Including more control variate references in Section 6: We will add more references in the final manuscript as long as we have enough space after revisions (if not we could provide a more exhaustive related work section in the appendix).
>
>
> (1) Titsias, Michalis, and Jiaxin Shi. "Double control variates for gradient estimation in discrete latent variable models." In International Conference on Artificial Intelligence and Statistics, pp. 6134-6151. PMLR, 2022.

---

> > ### Comment · Reviewer_SSK5 · 2023-08-11
> > **Further Response**
> >
> > Thank you for the clarifications. I am happy with the updated pseudocode addressed to Reviewer NBCj. Reorganizing the sections would make the paper much easier to read and free up some space (I see some redundancy from introducing/explaining the concept of control variates multiple times.) Nevertheless, I am now happy to increase my score.
> >
> > As per the naming issue, I have looked up some synonyms to "double" or "two" and found words like "duo" or "bi-". Or, given the paper addresses all of the stochasticity present in BBVI, maybe something like "total" or "complete" could also be considerable. Also, given that Titsias chose "double," I think my original suggestion of "doubly" is also fine as it is not exactly the same. Hope this is helpful!

---

### Official Review · Reviewer_UG9s · 2023-07-07

**Soundness:** 3 good
**Presentation:** 3 good
**Contribution:** 3 good
**Rating:** 5
**Confidence:** 4

**Summary:**

The paper presents a method for variance reduction in stochastic gradient estimation in doubly stochastic variational inference
where there exists two sources of variance: (i) Monte Carlo noise when sampling from the variational distribution and (ii) gradient
variance due to the minibatch sampling. The authors consider reparametrizable Gaussian variational inference and introduce a control variate that tries to reduce simultaneously both sources of variance. This control variate combines previous ideas, such as a second order Taylor expansion of the function as in [19], and it seems to reduce the variance in the presented experiments.



**Strengths:**

The paper is very clearly written and all derivations appear to be correct. It contains also intuitive discussions why the proposed "coupled" control variate can be useful as opposed to other control variates that deal separately with each source of variance.

Certainly variance reduction is a very important topic for stochastic gradient estimation in variational inference, and the paper proposes a potentially useful method.

The experiments provide many details including running times.

**Weaknesses:**

I was not so impressed by the experimental results for two reasons. Firstly the models used are quite small and it would be useful to include e.g. a big neural network. Secondly, I am not sure if the comparison is done in a fair way for methods such the "naive" method.  This is because the "dual method is more expensive and requires more gradient evaluations, such as the ones for the numerical approximation of the Hessian-vector products. Given that computations are dominated by the number of gradient evaluations, a fair comparison should try to match this number across different estimators. For example,  for the "naive" estimator someone could increase the minibatch size so that the number of gradient evaluations matches the one of the "dual" method.

**Questions:**

Why the control variate in equation (16) controls only the variance for the $\mu$?  Is this discussed in [9]?

**Limitations:**

The limitations are explained above regarding the current experimental comparison.

---

> ### Author Rebuttal · Authors · 2023-08-08
>
>
> Thank you for your careful review and suggestions.
>
> - Choice of models: We would like to clarify that the models we experiment with have rather high latent dimensionality. The models presented in Fig. 4 have 7840, 12544, and 5525 latent variables, respectively. Certainly, larger models exist, but these represent a real challenge for existing algorithms as evidenced by the performance of the baselines.
>
> - Fairness of comparison: Thanks for pointing this out. Following this suggestion, we conducted new experiments on MNIST—the results are shown in the PDF included as part of the general rebuttal. In these experiments, we use a larger batch size for naive and cv (300 and 200 respectively) to ensure all estimators have the same complexity per iteration. The dual estimator still shows better convergence than baseline estimators in this setup. We will include these results in the revised manuscript. It is worth pointing out that a primary goal of our original experiments was to illuminate the theory, making clear what the benefit of controlling variance is with the same (n,ε) values. But certainly, we agree that improvement in wall-clock speed is the ultimate goal. We partially addressed this by including the wall-clock time for each estimator in Table 2 and showing results in terms of wall-clock time in Figure 7 in the appendix (original submission). However, we agree that it is also useful to compare in terms of different minibatch sizes that hold the number of evaluations of ∇f per iteration constant, and we will include these results in the manuscript as well.
>
> - Controlling variance for $\mu$: Yes, this is discussed in [9]. Briefly, $\mu$ accounts for most of the gradient variance in mean-field VI, as is illustrated by the first row in Figure 1 in [9]. We empirically confirm this, and our experiments show that controlling the variance on $\mu$ is sufficient to significantly improve convergence speed.

---

> > ### Comment · Reviewer_UG9s · 2023-08-15
> > **Thank you for your reply**
> >
> > Thanks for adding the comparison by matching the number of gradient evaluations. This makes the experimental evaluation more complete.

---

### Official Review · Reviewer_11Uw · 2023-07-08

**Soundness:** 3 good
**Presentation:** 2 fair
**Contribution:** 2 fair
**Rating:** 5
**Confidence:** 1

**Summary:**

Existing stochastic methods for black-box variational inference only attempt to reduce the noise either due to data subsampling or Monte-Carlo sampling of the expectation. This paper proposed a new "dual control variate", which addresses both types of noise at the same time. In experiments, the proposed control variate is shown to perform favorably to existing baselines.

**Strengths:**

1) The paper disentangles the effects of noise through the data and noise from Monte-Carlo estimation of the expectation in the design of the new dual control variate, which addresses both at the same time.
2) The proposed control variate is shown to greatly improve performance on the considered examples, at seemingly minimal overhead. If the x-axis in the plots would have been wall clock time, this would be clearer to see.


**Weaknesses:**

The main weakness of the paper is in my opinion that the experimental evaluation is a bit lacking:
a) larger experiments would be desirable, see questions;
b) the plots were hard to read (difficult to distinguish the red and green curves), and could be wrt. wall clock time and not iterations.

The algorithm looks promising, and I like that it can be used in a "black-box" fashion with any optimizer (e.g. Adam); but I believe it is not ready yet in its current state and could require a more thorough experimental evaluation to warrant a clear acceptance.

**Questions:**

Would the proposed dual control variate approach also work for VI in neural networks? Experiments on modern neural networks (e.g. ResNets, transformers) could be a way to convincingly demonstrate the utility of the proposed method.

How much additional overhead (memory & runtime) does it have when compared to, say, Bayes-by-Backprop when actually implemented in practice?

From Table 2 it seems that one would require three backpropagation passes rather than a single one for the naive method -- it would be interesting to see whether this additional effort has a real practical benefit (e.g. higher accuracy, faster convergence wrt wall clock time).


**Limitations:**

All limitations are addressed.

---

> ### Author Rebuttal · Authors · 2023-08-08
>
> Thank you for your careful review and suggestions.
>
> - Experiments and models used: Our experiments focus on Bayesian inference on mechanistic models, a well-recognized area. Many of the models we consider are high-dimensional, with latent dimensions going from 5000 to 12000 for the larger models (lines 211 to 213), and represent complex problems for approximate inference algorithms. Regarding Bayes-by-backprop (BBB), it is essentially mean-field Gaussian VI applied to Bayesian neural networks using reparameterization gradients (our naive estimator).
>
> - Runtime: We plot convergence with respect to iterations for the sake of reproducibility because wall-clock measurements are sensitive to implementation. However, we provide the exact time per iteration for all methods in Table 2, which allows comparison. We also provide results directly showing convergence in terms of wall-clock time in Figure 7 (appendix)

---

> > ### Comment · Reviewer_11Uw · 2023-08-17
> >
> > Thanks for the clarification!

---

### Official Review · Reviewer_xUai · 2023-07-25

**Soundness:** 3 good
**Presentation:** 2 fair
**Contribution:** 3 good
**Rating:** 6
**Confidence:** 3

**Summary:**

The paper addresses the drawback of the black-box variational inference framework for Bayesian posterior inference by proposing dual control variate that is capable of jointly reducing the variances from both data subsampling and Monte Carlo sampling. The experimental evaluations on various datasets demonstrates reduced variance and improved optimization.

**Strengths:**

The paper is fairly written well. The background of the black-box variational inference (BBVI) is nicely described. The doubly-stochastic optimization problem in BBVI's gradient estimation is clearly explained involving two sources of randomness - Monte Carlo sampling from the variational posterior and data subsampling from the full dataset.

The proposed dual control variate that jointly controls Monte Carlo and subsampling noise in BBVI to create approximations of the target for each datum where the Monte Carlo noise can be integrated exactly, addressing both forms of noise and interactions between them.

Experimental evaluation and visualization of dual estimator.



**Weaknesses:**

- Although the idea is novel but its usefulness/impact could have explained well.

- The approximation function for the gradient estimators g_cv and g_dual could have been clarified.

- It is unclear how the noise in Monte Carlo sampling influences the noise in data sampling. Is there a way to measure it?

- What is the role of Beta in eqn(14 - 15). Is it experimentally evaluated?



**Questions:**

I am not fully familiar with this subject. However, I did get idea about the problem and how the propose idea could address it. Therefore, I  will consider the reviews of other reviewers.

The main problem is the writing/explanation of the technical terms to make readers to understand the problem. The paper started well and somehow there is a lost connection between experimental evaluation to demonstrate the impact of the dual control variate real-world applications

**Limitations:**

There is no issue with potential negative societal impact. However, the article does not address limitations (if any).

---

> ### Author Rebuttal · Authors · 2023-08-08
>
>
> Thank you for your careful review and suggestions.
>
> - Presentation/clarity: We agree that the presentation could be improved. We will aim to better clarify technical terms and impact, as well as incorporate suggestions from reviewers SSK5 and NBCj.
>
> - Approximation function: We appreciate that the current way the paper talks about the approximation function $\tilde{f}$ is not ideal. Looking at the paper closely, we see that we currently only very briefly mention in Section 4 that a Taylor expansion is a common choice, but we do not provide any details until Section 7. We agree it would be better to be more concrete earlier and will work to improve the manuscript in this respect.
>
> - Monte Carlo and subsampling noise: When estimating the gradient, these two random variables (n) and (ε) are sampled independently. However, the amount of Monte Carlo noise will typically differ by minibatch. We attempt to measure the interaction in Figure 1 and Table 1 where ∇f(w;n) represents the gradient estimator with the Monte Carlo variable ε integrated out and ∇f(w;ε) represents the gradient estimator with the data subsampling variable n integrated out. We see that results vary from problem to problem (and by iteration inside each problem). However, at a high level, we see that interactions do exist, in the sense that the variance adds "superlinearly". More precisely, the variance with both sources of noise ($V_{ε, n} ∇f(w;n,ε)$) is larger than the sum of the two independent sources of noise ($V_ε ∇f(w;ε) + V_n ∇f(w;n)$)
>
> - What is beta: It is the mixture weight of $c_\text{cv}$ and $c_\text{inc}$ for the ensembles approach from Section 5.3. We do not experimentally evaluate this approach because computing $c_\text{inc}$ is intractable in general (see line 158-164 for more details). We will emphasize this point in Sec 5.3 in the revised manuscript.

---

> > ### Comment · Reviewer_xUai · 2023-08-16
> > **Thank you for the clarifications**
> >
> > Thanks for addressing my concerns.
> >
> > Don't forget to improve the content organization and presentation clarity in the revised paper.

---

> > > ### Author Response · Authors · 2023-08-16
> > >
> > > Thank you for the response! We will absolutely make these changes to the organization and presentation clarity—we agree that these would increase the impact of the paper.

---

### Author Rebuttal · Authors · 2023-08-08

We would like to thank the reviewers for their careful reviews.

In the PDF file, we provided the following additional content:

- UG9s has concern regarding the fairness of the experiments, in particular, since dual requires extra gradient calls at each iteration, the baseline estimators should use a larger batch size correspondingly to ensure different estimators have the same gradient call budget at each iteration. We have therefore included new experiments on MNIST (Figure. 1) where we used different batch sizes for naive, cv, and dual: Dual is 3 times more expensive than naive and 1.5 times more expensive than cv (in terms of gradient call), therefore we use a batch size of 300, 150, and 100 for naive, cv and dual respectively such that they have the same cost per iteration. Under this condition, we still observe dual performs better than baseline methods.

- Reviewer NBCj raises concerns about the O(ND) memory cost of dual caused by the underlying SAGA, we agree that this is a potential limitation. In the PDF file, we present the results (Figure. 2) for the SVRG version of dual control variate on Australian and we observe performance close to the SAGA version. The SVRG version bypasses the additional memory cost by introducing an outer loop that computes the full gradient of the control variate at every $m$ iteration and uses it as the expectation with respect to $n$. This variation of dual does not require additional memory cost but would cost extra gradient evaluation and introduce an extra hyperparameter. In our experiments, we compute the full gradient every 5 epochs, equivalent to 0.2 additional gradient calls per iteration.

---

### Decision · Program_Chairs · 2023-09-21

**Decision:**

Reject

**Comment:**

This paper presents an idea involving dual control variates for expediting black-box variational inference. The concept appears to be novel and interesting, and the paper is generally well-written. However, after thorough discussions, I would recommend rejecting the paper in its current form mainly due to its limited evaluations and the inconsistency of its claims.

The paper begins with the general idea of a new method to reduce gradient variance in Bayesian posterior inference from data subsampling and Monte Carlo sampling. I believe this could be a valuable contribution to the field. However, the paper's evaluation primarily focuses on linear models, including Bayesian (multi-class) logistic regression, PCA, and the Bradley Terry model. Even if we exclude the more "modern/larger" Bayesian neural networks, the exclusive focus on linear models limits the impact of the proposed method. This also contradicts the paper's claim to be a general improvement of BBVI. If the primary focus is on the application of linear models, then this should be explicitly stated in the paper and compared against other Bayesian posterior methods tailored for linear models, such as variational Bayesian logistic regression or variational PCA.